# Multiple Novel Human Norovirus Recombinants Identified in Wastewater in Pretoria, South Africa by Next-Generation Sequencing

**DOI:** 10.3390/v14122732

**Published:** 2022-12-07

**Authors:** Victor Vusi Mabasa, Walda Brenda van Zyl, Arshad Ismail, Mushal Allam, Maureen Beatrice Taylor, Janet Mans

**Affiliations:** 1Department of Medical Virology, Faculty of Health Sciences, University of Pretoria, Private Bag X323, Gezina, Pretoria 0031, South Africa; 2National Health Laboratory Service, Tshwane Academic Division, Pretoria 0002, South Africa; 3Sequencing Core Facility, National Institute for Communicable Diseases, National Health Laboratory Service, Johannesburg 2192, South Africa; 4Department of Biochemistry and Microbiology, Faculty of Science, Engineering and Agriculture, University of Venda, Thohoyandou 0950, South Africa; 5Department of Genetics and Genomics, College of Medicine and Health Sciences, United Arab Emirates University, Al Ain 15551, United Arab Emirates

**Keywords:** norovirus, recombinants, South Africa, wastewater, next-generation sequencing, Tshwane, Pretoria

## Abstract

The genogroup II genotype 4 (GII.4) noroviruses are a major cause of viral gastroenteritis. Since the emergence of the Sydney_2012 variant, no novel norovirus GII.4 variants have been reported. The high diversity of noroviruses and periodic emergence of novel strains necessitates continuous global surveillance. The aim of this study was to assess the diversity of noroviruses in selected wastewater samples from Pretoria, South Africa (SA) using amplicon-based next-generation sequencing (NGS). Between June 2018 and August 2020, 200 raw sewage and final effluent samples were collected fortnightly from two wastewater treatment plants in Pretoria. Viruses were recovered using skimmed milk flocculation and glass wool adsorption-elution virus recovery methods and screened for noroviruses using a one-step real-time reverse-transcription PCR (RT-PCR). The norovirus BC genotyping region (570–579 bp) was amplified from detected norovirus strains and subjected to Illumina MiSeq NGS. Noroviruses were detected in 81% (162/200) of samples. The majority (89%, 89/100) of raw sewage samples were positive for at least one norovirus, compared with 73% (73/100) of final effluent samples. Overall, a total of 89 different GI and GII RdRp-capsid combinations were identified, including 51 putative novel recombinants, 34 previously reported RdRp-capsid combinations, one emerging novel recombinant and three Sanger-sequencing confirmed novel recombinants.

## 1. Introduction

Human noroviruses are the most common cause of epidemic and sporadic acute gastroenteritis in all age groups globally [1]. An estimated 212,489 deaths occur every year worldwide due to norovirus-associated gastroenteritis [2] and many of these deaths are in lower- to middle-income countries (LMICs) [3]. Although individuals of all ages are affected, children under the age of five [4], the elderly [5], and the immunocompromised are prone to severe illness [6]. These highly infectious viruses are transmitted faecal-orally through person-to-person contact and the consumption of contaminated food or water [7,8]. Hence, most norovirus outbreaks are found in closed settings such as nursing homes or linked to shared food and water sources at sporting and entertainment events, cruise ships, and airlines [9]. There is no licensed vaccine nor specific antiviral drug to prevent and treat disease caused by human noroviruses [10,11]. 

Noroviruses are non-enveloped and have a linear, non-segmented, single-stranded, positive-sense RNA genome of ~7.7 kb [12]. They are classified under the genus *Norovirus* in the *Caliciviridae* family together with ten other genera that infect a wide range of hosts [13]. The *Norovirus* genus is divided into ten genogroups (GI-GX), which are subdivided into 60 P-types and 49 genotypes based on the nucleotide diversity of the RNA-dependent RNA polymerase (RdRp) and the amino acid diversity of the complete major capsid (VP1) gene, respectively [14]. Genogroups I, II, IV, VIII, and IX comprise at least 49 P-types and 35 genotypes that are known to cause disease in humans [14]. Despite the vast diversity within these genogroups, genotype GII.4 viruses have been the most prevalent cause of norovirus-associated gastroenteritis [15]. Their success is largely driven by the periodic emergence (every 2 to 5 years) of novel strains and intergenotype recombination that often results in a surge of infections globally [16]. However, there has not been a pandemic caused by a novel GII.4 variant since the emergence of Sydney_2012 [17]. The GII.4 strains that are currently circulating have a Sydney_2012 capsid with different P-types due to intergenotype recombination [18]. The emergence and dominance of novel GII.17 strains between 2014 and 2016 in different parts of the world has placed an emphasis on the importance of continuous surveillance for human noroviruses at population level [19]. Norovirus surveillance from raw sewage has proved to be feasible in determining the diversity of viruses circulating in a given population [19,20,21]. Unlike clinical surveillance, environmental surveillance provides a broad insight into the epidemiology and distribution of enteric viruses as it includes strains from asymptomatic individuals and those with mild disease that shed the virus but do not seek medical help [22]. Next-generation sequencing (NGS) is a versatile tool utilised in virus genomic diversity studies as large amounts of genetic material can be sequenced simultaneously, at a low cost [23]. In this study, Illumina MiSeq was used to determine the diversity of human noroviruses from raw sewage and final effluent water samples in selected wastewater treatment works (WWTWs) in Pretoria, South Africa.

## 2. Materials and Methods

### 2.1. Sample Collection and Virus Recovery

Between June 2018 and August 2020, raw sewage (1 L) and final effluent (10 L) water samples were collected bi-weekly from two WWTWs in Pretoria, in the City of Tshwane Metropolitan Municipality, South Africa. All samples were collected between 7 and 9 am during the peak morning flow. The WWTWs receive sewage from communities of approximately 1.42 million people and use activated sludge and chlorine treatment [24]. A modified pre-flocculated skimmed milk (PSM) methodology was used to recover viruses from the raw sewage samples [25]. Briefly, the pH of raw sewage samples was adjusted to 3.5 when 1 N HCl (Merck KGaA, Darmstadt, Germany) or 1 M NaOH (Merck) was added. The conductivity of water was increased by adding 33.3 g of sea salt per sample. This was performed to ensure the formation of flocculated material. A 1% (*w*/*v*) PSM solution was prepared by dissolving 1 g skim milk powder (Oxoid Ltd., Basingstoke, United Kingdom) in 100 mL autoclaved distilled water and the pH was adjusted to 3.5 with 1 N HCl (Merck). The prepared PSM was immediately added to the conditioned raw sewage sample and left to stir at room temperature for 12 hours to allow viruses to adsorb to the flocculated material. The mixture was centrifuged (Sorvall^®^ Super T20, du Pont, Wilmington, DE, USA) at 10,000× *g* for 30 min at 4 °C. The supernatant was discarded, and the pellet was resuspended in 10 mL phosphate buffered saline (PBS, pH 7.5 Sigma-Aldrich, St. Louis, MO, USA), treated with 0.2 volumes of chloroform (Merck) and centrifuged at 10,000× *g* for 5 min at 4 °C. The final effluent samples were processed using the modified glass wool adsorption-elution technique as described previously (Mabasa et al., 2018). All virus concentrates were stored as 1 mL aliquots at −20 °C.

### 2.2. Nucleic Acid Extraction and Virus Detection

Before nucleic acid extraction, all virus concentrates were re-treated with chloroform to further remove PCR inhibitors. Briefly, 200 µL chloroform (Merck) was added to 1 mL virus concentrate and vortexed vigorously for 1 min followed by centrifugation at 5000× *g* for 5 min at room temperature. The aqueous phase was transferred to a clean 2 mL microcentrifuge tube and spiked with a known concentration of murine mengovirus strain MCo, which served as an extraction efficiency control [26]. Total nucleic acid was extracted using a QIAamp UltraSens Virus kit (Qiagen, Valencia, CA, USA) according to manufacturer’s instructions. The nucleic acid was eluted in 100 µL nuclease-free water and stored as 10 µL aliquots at −80 °C. A one-step real-time RT-PCR was used for the detection of norovirus GI, GII, and mengovirus genomic material using a QuantiFast Pathogen RT-PCR + IC Kit (Qiagen) with published Taqman probes and primers (Thermo Scientific, Waltham, MA, USA) (Table 1). All assays were performed on the QuantStudio™ 5 real-time system (Applied Biosystems, Foster City, CA, USA). The kit’s internal control was used to monitor the efficiency of target amplification. A cycle threshold (Ct) value of 40 was used as a cut-off for norovirus-positive samples and ≥1% extraction efficiency was regarded as a successful extraction of nucleic acid from the samples as described in ISO 15216-1 [27].

### 2.3. Molecular Typing of Noroviruses

A two-step RT-PCR was used to determine P-types and genotypes of detected noroviruses using EmeraldAmp^®^ Max HS PCR Master Mix (Takara, Shiga, Japan) with published primers (Table 1). These target the 579 bp and 570 bp fragments representing the BC region of GI and GII genomes, respectively [28]. Complementary DNA was synthesised using 10 µL of the extracted nucleic acids, 30 µM random hexamers (Thermo Scientific), 0.5 mM dNTPs (New England Biolabs Inc., Ipswich, MA, USA), 1 × ProtoScript^®^ II reverse transcriptase reaction buffer (New England Biolabs Inc.), 0.1 M DTT (New England Biolabs Inc.), 20 U RNase inhibitor (New England Biolabs Inc.), and 200 U ProtoScript^®^ II reverse transcriptase (New England Biolabs Inc.). Subsequently, 5 µL of the generated cDNA was used as template for first-round amplification of the BC region. A second-round semi-nested PCR was performed using 1 µL of first-round PCR as a template. This was performed to increase the amplicon yield and to incorporate Illumina MiSeq overhang adapters (Table 1) to the amplicons. The same amplification conditions were used for the first- and second-round PCR (Table 1). The resulting amplicon fragments were analysed on 1.5% agarose gels, purified using Agencourt AMPure XP beads (Beckman Coulter, Brea, CA, USA) following the manufacturer’s instructions and sent to the National Institute for Communicable Diseases (NICD), Johannesburg, SA for NGS.

**Table 1 viruses-14-02732-t001:** Real-time RT-PCR and typing PCR assay primers and probes.

**Virus**	**Primer/Probe**	**Sequence (5′–3′) ***	**Polarity**	**Location**	**Amplification Conditions**	
Norovirus GI	QNIF4 ^a^	CGCTGGATGCGNTTCCAT	+	5291–5308 †	50 °C, 20 min 95 °C, 5 min **×45 cycles** 95 °C, 15 s 55 °C, 30 s 65 °C, 30 s	
	NV1LCR ^b^	CCTTAGACGCCATCATCATTTAC	−	5354–5376 †		
	NVGG1 ^b^	FAM-TGGACAGGAGAYCGCRATCT-TAMRA	+	5321–5340 †		
Norovirus GII	QNIF2 ^c^	ATGTTCAGRTGGATGAGRTTCTCWGA	+	5012–5037 *	50 °C, 20 min 95 °C, 5 min **×45 cycles** 95 °C, 15 s 60 °C, 30 s 65 °C, 30 s	
	COG2R ^d^	TCGACGCCATCTTCATTCACA	−	5080–5100 *		
	QNIFS ^c^	FAM -AGCACGTGGGAGGGCGATCG-TAMRA	+	5042–5061 *		
Mengovirus	Mengo110F ^e^	GCGGGTCCTGCCGAAAGT	+	110–127 ❄	50 °C, 20 min 95 °C, 5 min **×45 cycles** 95 °C, 15 s 60 °C, 30 s 65 °C, 30 s	
	Mengo209R ^e^	GAAGTAACATATAGACAGACGCACAC	−	245–270 ❄		
	Mengo147 ^e^	MGB-ATCACATTACTGGCCGAAGC-TAMRA	+	208–227 ❄		
**Target**	**Primer**	**Sequence (5′–3′) ***	**Polarity**	**Location**	**Amplification conditions**	
					**1st round**	**2nd round**
Norovirus GI BC region	JV12Y ^f^	ATACCACTATGATGCAGAYTA	**+**	4552–4572 †	**×40 cycles** 95 °C, 15 s 50 °C, 30 s 72 °C, 1 min 72 °C, 5 min	**×40 cycles** 95 °C, 15 s 50 °C, 30 s 72 °C, 45 s 72 °C, 5 min
	MON432Tx ^g^	**TCGTCGGCAGCGTCAGATGTGTATAAGAGACAG**TGGACICGYGGICCYAAYCA	**+**	5093–5112 †		
	G1SKRTx ^h^	**GTCTCGTGGGCTCGGAGATGTGTATAAGAGACAG**CCAACCCARCCATTRTACA	**−**	5653–5671 †		
Norovirus GII BC region	JV12Y ^f^	ATACCACTATGATGCAGAYTA	**+**	4279–4299 *	**×40 cycles** 95 °C, 15 s 55 °C, 30 s 72 °C, 1 min 72 °C, 5 min	**×40 cycles** 95 °C, 15 s 55 °C, 30 s 72 °C, 45 s 72 °C, 5 min
	MON431Tx ^g^	**TCGTCGGCAGCGTCAGATGTGTATAAGAGACAG**TGGACIAGRGGICCYAAYCA	**+**	4821–4840 *		
	G2SKRTx ^h^	**GTCTCGTGGGCTCGGAGATGTGTATAAGAGACAG**CCRCCNGCATRHCCRTTRTACAT	**−**	5367–5389 *		
Norovirus GI BC region for viruses with P7 P-type	JV12Y ^f^	ATACCACTATGATGCAGAYTA	+	4552–4572 †	**×40 cycles** 95 °C, 15 s 50 °C, 30 s 72 °C, 1 min 72 °C, 5 min	**×40 cycles** 95 °C, 15 s 50 °C, 30 s 72 °C, 1 min 72 °C, 5 min
	GIP7F ^i^	CTGATGTGGAATTTGACCCAATAAAG	**+**	4890–4915 †		
	G1SKR ^h^	CCAACCCARCCATTRTACA	**−**	5653–5671 †		

^a^ [29], ^b^ [30], ^c^ [31], ^d^ [32], ^e^ [33], ^f^ [34], ^g^ [35], ^h^ [36], ^i^ (in-house designed). * Degeneracy code: Y = C/T; R = A/G; W = A/T; I = Inosine; N = any base. Bold bases = MiSeq adapter. Probe labels: 6-carboxy fluorescein (FAM), minor groove binder (MGB), and 6-carboxy-tetramethylrhodamine (TAMRA). Location based on corresponding nucleotide position of M87661 (†), X86557 (*) and L22089 (❄).

### 2.4. DNA Library Preparation and Illumina Sequencing

Amplicons were pooled equimolar based on wastewater sample type and month of collection. For the amplicon library preparations, amplicons were indexed using IDT for Illumina Nextera DNA Unique Dual Indexes (Illumina Inc., San Diego, CA, USA) according to manufacturer’s recommendation. The amplicon library concentration was measured on the Qubit^®^ 3.0 Fluorometer (Invitrogen) and the size of the amplicons were then visualised using the 4200 TapeStation (Agilent Technologies, Santa Clara, CA, USA). A negative control was used in the amplicon preparation step. No amplification was observed in the negative control. The purified amplicon libraries were normalised to 2 nM and then sequenced on the Illumina MiSeq platform using the Illumina MiSeq v3 kit (Illumina Inc.), to obtain 2 × 300 bp paired-end sequences.

### 2.5. NGS Data Analysis

The paired-end Illumina MiSeq reads were received as FASTQ files, which were merged using default settings and format changed to FASTA using BBMap [37]. The merged reads were length-filtered to eliminate sequences that were below 570 bases using Galaxy [38]. The reads were clustered based on a 90% sequence similarity threshold using CD-HIT-EST [39] and a FASTA file of representative sequences was generated. The proportion representation of each genotype was then calculated by dividing the number of merged reads attributed to the genotype over the total number of merged reads. The number of sequences clustered to each representative genotype was used to quantify the abundance of each genotype within wastewater samples.

### 2.6. Sanger Sequencing

A two-step RT-PCR was performed for the amplification and characterisation of putative novel recombinants using EmeraldAmp^®^ Max HS PCR Master Mix (Takara) with in-house designed and published primers [36] (Inqaba Biotech™, Pretoria, South Africa). A total of 100 norovirus GI.P7 sequences were retrieved from NCBI GenBank and aligned using BioEdit [40] in order to identify conserved regions for primer design. Complementary DNA was synthesised as described above. Two rounds of PCR were performed using the same reaction components and amplification conditions to increase amplicon yield (Table 1). Amplicons were analysed on 1.5% agarose gels and purified with Agencourt AMPure XP beads. Subsequently, the amplicons were cloned using the ClonJET^TM^ PCR cloning kit (Thermo Scientific) and transformed into Lucigen E. cloni 10G competent cells (Lucigen Corp., Middleton, WI, USA). Ten randomly selected clones were subjected to colony PCR to confirm the presence of the insert fragment. A 20 µl reaction mixture comprised of 1 × OneTaq^®^ Quick-load^®^ Master Mix (New England Biolabs Inc.), 0.2 µM forward and reverse pJET1.2 primers (Thermo Scientific) and nuclease-free water. The PCR products were analysed on 1.5% agarose gels and clones with the correct insert fragment size were purified with a DNA Clean and Concentrator™-25 Kit (Zymo Research, Irvine, CA, USA). The clones were sequenced using pJET1.2 primers and the ABI PRISM BigDye1 Terminator version 3.1 Cycle Sequencing kit on an ABI 3130 automated analyser (Applied Biosystems).

### 2.7. Norovirus Phylogeny and Recombination Analysis

Phylogenetic analyses of the detected strains were performed on the partial polymerase (265 bp) and capsid (314 bp) sequences to determine norovirus P-types and genotypes, respectively. Representative sequences were aligned using MAFFT version 7 [41] and headers edited with FaBox version 1.5 [42] to match our study identification codes. The sequences were then analysed with the online Norovirus Typing Tool version 2.0 [43] to assign polymerase and capsid types. BLAST-n (http://blast.ncbi.nlm.nih.gov/Blast.cgi - accessed between October 2018 and July 2022) was used to compare the study sequences to norovirus sequences in GenBank and putative recombination breakpoint analysis was performed using SimPlot version 3.5.1 [44]. The analysis was carried out using a Kimura 2-parameter distance model [45] and a window size of 200 bp and a step of 20 bp. Phylogenetic trees were constructed using the Maximum Likelihood method in MEGA X supported by 1000 bootstrap replicates and the evolutionary distances were determined using the Kimura 2-parameter model. Nucleotide sequences obtained from the samples were submitted to NCBI GenBank (accession numbers: OP218002-OP218007, OP236435-OP236446, OP245964-OP245995, OP246120-OP246202, OP246242-OP246373, OP289691-OP289693, OP289784-OP289798, OP295607-OP295694, OP297036-OP297041, OP297403-OP297417, OP303353-OP303356, OP303495-OP303519, OP307860-OP310020, and OP746072-OP747040). The sequences were labelled according to month and year of detection, wastewater type (R = raw sewage, F = final effluent), genotype and P-type, e.g., 618RGII.2[P2] = GII.2[P2] virus detected in raw sewage collected in June 2018.

### 2.8. Statistical Analyses

Descriptive statistics were used to summarise norovirus detection rates and proportions of genotypes. The detection rates of norovirus GI and GII in raw sewage and final effluent were compared using the Mantel-Haenszel chi square test. (http://www.openepi.com/TwobyTwo/TwobyTwo.html, accessed on 17 November 2022).

## 3. Results

### 3.1. Detection of Noroviruses

A total of 200 wastewater samples were collected during the study period: 50 raw sewage and 50 final effluent samples from each WWTW. Samples could not be collected between the 2nd week of March 2020 and the end of June 2020 due to the COVID-19 pandemic lockdown restrictions. Overall, noroviruses were detected in 81% (162/200) of wastewater samples, comprised of 89/100 raw sewage and 73/100 final effluent. Genogroup I viruses were detected in 40% (80/200) and GII dominated being detected in 78% (156/200) of the samples. Both GI and GII viruses were detected more frequently in raw sewage (GI—*p* < 0.001; GII—*p* < 0.02) compared with final effluent. The majority (80%; 64/80) of GI viruses were detected in raw sewage and 20% (16/80) in final effluent, whereas GII was more evenly distributed between raw sewage (54%; 85/156) and final effluent (46%; 71/156) (Figure 1). The kit’s internal positive control (QuantiFast Pathogen RT-PCR) was detected in 100% of the wastewater samples tested at the required Ct value; therefore, no re-testing was required. Mengovirus was detected in all samples and concentrations varied between samples; however, the extraction efficiencies were above 1%.

### 3.2. Molecular Typing of Noroviruses

The BC region of norovirus genomes was successfully amplified from viruses in 56.3% (45/80) and 56.4% (88/156) of norovirus GI and GII positive samples, respectively. This resulted in 40 (23 raw sewage and 17 final effluent) NGS amplicon pools. Thirteen GI P-types were identified in wastewater and these included GI.P7 (43%; 17/40), GI.P13 (38%; 15/40), GI.P3 (28%; 11/40), GI.P4 (23%; 9/40), GI.P10 (23%; 9/40), GI.P11 (15%; 6/40), GI.P9 (8%; 3/40), GI.P1, GI.P2, GI.P5, GI.P6, GI.P8, and GI.P12 (5%; 2/40 each). These P-types circulated together with nine genotypes, which included GI.3 (50%; 20/40), GI.7 (48%; 19/40), GI.4 (23%; 9/40), GI.6 (18%; 7/40), GI.8 (10%; 4/40), GI.5 (8%; 3/40), GI.9 (8%; 3/40), GI.1 (5%; 2/40), and GI.2 (5%; 2/40). Overall, a total of 44 different GI RdRp-capsid combinations were identified and these included 28 putative novel recombinants, 13 previously reported RdRp-capsid combinations and three Sanger-sequencing confirmed novel recombinants (Figure 2A,B, Appendix A). The top five frequently detected GI viruses were GI.7[P7] (48%; 19/40), GI.3[P13] (40%; 16/40), GI.3[P3] (28%; 11/40), GI.3[P7] (25%; 10/40), and GI.4[P4] (23%; 9/40). Although GI.7[P7] was detected at least once in every season, there was a noticeable peak of these viruses in the second half of the year with most of them identified in winter and spring (Figure 2A,B). The GI.3[P13] viruses were identified in small proportions in raw sewage between 2018 and 2019. However, in 2020 they dominated both in raw sewage and final effluent samples (Figure 2A,B). The in-house-designed forward primer (GIP7F) successfully amplified a 770 bp BC region fragment of GI viruses with a P7 RdRp. The viruses identified included GI.7[P7] and three (GI.1[P7], GI.3[P7], GI.4[P7]) of the putative novel recombinants. For GII, nine P-types were identified and these included GII.P7 (65%; 26/40), GII.P16 (60%; 24/40), GII.PNA (30%; 12/40), GII.P17 (20%; 8/40), GII.P31 (18%; 7/40), GII.P33 (10%; 4/40), GII.PNA7 (10%; 4/40), GII.P8 (5%; 2/40), and GII.P21 (3%; 1/40). These P-types circulated together with 13 genotypes, which included GII.2 (55%; 22/40), GII.9 (48%; 19/40), GII.4 (40%; 16/40), GII.6 (35%; 14/40), GII.17 (20%; 8/40), GII.3 (13%; 5/40), GII.12 (13%; 5/40), GII.7 (10%; 4/40), GII.1 (8%; 3/40), GII.8 (5%; 2/40), GII.10 (5%; 2/40), GII.13 (3%; 1/40), and GII.16 (3%; 1/40). Overall, 45 different GII RdRp-capsid combinations were found in circulation, and these included 23 putative novel recombinants, 21 previously reported RdRp-capsid combinations and one emerging novel recombinant (GII.12[P16]) (Figure 3A,B, Appendix A). The predominant GII viruses identified were GII.2[P16] (55%; 22/40), GII.9[P7] (48%; 19/40), GII.6[P7] (35%; 14/40), GII.2[P7] (30%; 12/40), and GII.9[P16] (20%; 8/40). The majority of the GII.2[P16] viruses were identified in samples collected in 2019 (Figure 3A,B). Almost all (95%; 20/21) of the GII.4 viruses were identified in samples collected between June and December. As many as 24 GI and 19 GII viruses were characterised from a single raw sewage sample in February 2020 and September 2019, respectively. The majority of untypeable viruses were in final effluent samples as shown by the blank spaces in Figure 2B and Figure 3B.

### 3.3. Recombination Analysis

The breakpoints of putative and confirmed novel recombinant noroviruses were identified from 106 representative GI (50) and GII (56) norovirus sequences (Appendix A). Single capsid/RdRp combinations were identified for 4 viruses, with the remaining 102 viruses exhibiting capsid/RdRp combinations that ranged from one capsid combined with 2 RdRps to 7 RdRps. Differences in breakpoints within the same cap-sid/RdRp combination further increased the diversity (up to seven strains with different breakpoints in the GI.3[P7] recombinant). In predominance order, the GI capsids included GI.3, GI.4, GI.6, GI.7, GI.1, GI.9, GI.5, GI.8, and GI.2. The GI.3 viruses recombined with seven different RdRps resulting in 19 strains making it the most diverse group of recombinant GI viruses. The least diverse recombinant groups were GI.5 and GI.8, with two different RdRps each, and GI.2 with one RdRp. The most common GI P-types identified in recombination were GI.P7 and GI.P13 accounting for almost a quarter (12/50) of strains, each. The most common GII recombinant strains were GII.2 and GII.9 with three different RdRps each and this resulted in 13 and 12 strains, respectively. The least diverse recombinant viruses harboured GII.13 and GII.16 capsids with one P-type resulting in one strain, each. The predominant GII P-type was GII.P7 followed by GII.P16, both accountable for 71% (40/56) of GII recombinant strains. The GII.P7 recombined with six different capsids resulting in 26 strains whereas GII.P16 was found in combination with four different capsids resulting in 14 strains. The ORF2 starting point was at nucleotide 266 for all GI and GII strains except for strain 1018RGI.1[P3] that started at position 270. The majority (62%; 66/106) of breakpoints were identified within the capsid gene and the remaining 38% were within the RdRp region. In addition, 76% of all breakpoints were between nucleotides 246 and 289.

### 3.4. Phylogenetic Analysis

The Maximum Likelihood phylogenetic trees of the GI partial RdRp and capsid were constructed to determine their relatedness to previously reported GI strains with sequences submitted to GenBank (Figure 4 and Figure 5). The analyses revealed relatedness between strains of the same P-type associated with different genotypes (Figure 4); however, there is a significant diversity within the capsids (Figure 5). Identified GI.3 viruses form two main clusters that further divide into several sub-clusters. Interestingly, the analysis also showed that the GI.3 Cluster I is more closely related to GI.8 than it is to GI.3 Cluster II. Likewise, GI.5 and GI.6 viruses form distinctive clusters and sub-clusters (Figure 5). The GI.9 viruses also form different clusters, but they are not closely related to their reference strains (Figure 5). Figure 6 and Figure 7 show the relationship between identified GII viruses and sequences available in GenBank. P-types GII.P7 and GII.P16 recombined with more genotypes compared with other identified P-types. In addition, the GII.PNA7 sequences from the study form two distinctive clusters, which group apart from the reference strain LC342059. The GII.3, GII.7, GII.9, GII.12, and GII.16 viruses clustered separately from their reference and closely related strains (Figure 7). Figure 8 displays the genetic diversity of the dominant GII.2, the well-documented GII.4, and previously dominant GII.17.

## 4. Discussion

To the best of our knowledge, this is the first study utilising NGS to understand the diversity and distribution of human norovirus circulating in wastewater samples in Africa. The use of NGS in norovirus surveillance provides an in-depth understanding of strains circulating at population level, at any given time. In addition, environmental NGS data reveal rare and emerging strains that might be missed with the cloning and Sanger sequencing approach. In this study, Illumina MiSeq was used to assess the norovirus GI and GII diversity in wastewater from Pretoria, SA. Overall, noroviruses were detected in 81% of the wastewater samples tested. As expected, the majority of these samples were raw sewage, which represented pooled stools from millions of individuals from the city. The treated final effluent samples also had detectable norovirus genomic material but less prevalent compared with raw sewage samples. No real-time RT-PCR inhibition was observed as the internal positive control was successfully detected at the required Ct value in all samples. A recent study from Botswana reported comparable results with norovirus GI detected in 67% and GII in 78% of raw sewage samples [47].

The data show that GII (78%) was more prevalent in the environment compared with GI (40%) and these findings mirror previous reports from Africa [19,22,48], Brazil [20], Thailand [49], and Sweden [50]. In addition, Figure 1B shows a decline in detection rate of GI virus genomes in final effluent samples. However, the norovirus GII detection rate was comparable in raw sewage and final effluent (Figure 1A,B). This suggests that GI viruses might be less stable in the environment compared with GII and thus more susceptible to wastewater treatment methods used. This is in contrast to findings reported by Sano and colleagues that the estimated log reductions in norovirus GI (1.48; 95% CI, 0.96–2.00) and GII (1.35; 95% CI, 0.52–2.18) were comparable when the conventional activated sludge treatment method was used [51]. However, differences in the stability of GI and GII have been described by Cromeans and co-workers, who reported that norovirus GI was more sensitive to reduction in RNA levels due to treatment with alcohols, chlorine, and high hydrostatic pressure than GII noroviruses [52]. The reduction in virions with intact capsids can lead to a lack of adequate, good quality nucleic acids. This can explain the lower numbers of merged reads (Appendix A) and fewer GI genotypes from final effluent samples (Figure 2B) in 2018 and 2019 compared with higher numbers of GII merged reads and increased virus diversity (Figure 3B). The overall molecular typing success rate was 56.4% for both norovirus GI and GII. The majority of the untypeable strains were detected in final effluent samples and this places emphasis on the point that lack of adequate good quality RNA template can be one of the reasons for low typing success rate. Despite this, NGS resulted in excellent data, revealing at least 89 different RdRp-capsid norovirus combinations of which the majority were putative novel recombinants. This is the first study to report such diverse data on circulating noroviruses and putative novel recombinants from environmental surveillance. Although these putative recombinants were identified in up to ten different amplicon pools, their merged reads proportion was less compared with that attributable to well-documented strains (Appendix A). This is possibly due to low virion concentration in the wastewater or primer-mismatch during PCR resulting in a poor representation of a strain in a pool of amplicons. Therefore, it would be rather challenging to identify such strains using the classical cloning and Sanger sequencing approach. Similar to other reports where norovirus diversity was studied in wastewater, this study proved the advantages of using NGS over Sanger sequencing. The GI.P7 RdRp dominated amongst other GI P-types and this correlates with recent data that show its prevalence as one the most reported norovirus P-types in public databases [53]. In this study, it was linked to the majority of the putative novel recombinants, and the viruses that carried it varied greatly (Figure 5). This prompted the design of GIP7F that was used together with a G1SKR reverse primer to amplify genomic material of viruses with GI.P7 RdRp for further phylogenetic analysis. This PCR confirmed the presence of the well-documented GI.7[P7] and three novel recombinants; GI.1[P7], GI.3[P7], and GI.4[P7] in circulation. Another interesting trend observed was the noticeable peak and dominance of GI.7[P7] viruses in winter and spring months. The GI.P7 strains associated with GI.4 capsid, and strains 1118RGI.7[P7], 219RGI.3[P7], and 619RGI.3[P7] cluster separately from each other and away from the remaining majority of GI.P7 strains associated with GI.3 and GI.7 capsids. The latter cluster includes GI.7 reference strain and closely related (93–99% nucleotide similarity) strains from paediatric patients in Brazil [54]. The GI.3[P13] viruses were first identified as a minority in raw sewage samples in July 2018 and this continued until their significant peak in October 2019 and continued this trend into 2020, where they were detected both in raw sewage and final effluent (Figure 2A,B). The GI.3 capsid was the most prevalent (50%; 20/40) among GI viruses and phylogenetically they are closely related (97–99% nucleotide similarity) to predominant GI.3 viruses that caused outbreaks in Taiwan [55]. Furthermore, these viruses formed two distinct clusters; GI.3 Cluster I and Cluster II (Figure 5) and the former was closer to GI.8 viruses than it was to GI.3 Cluster II by evolutionary distance. The GI.3 Cluster I was further divided into three sub-clusters and this vast diversity within GI.3 viruses is well-documented [56,57,58]. The GI.3 viruses have been reported amongst most prevalent noroviruses causing outbreaks and sporadic acute infections in Spain, Australia [59,60], Taiwan [55] and New Zealand [60]. This strengthens its place in the top 10 most frequently reported norovirus genotypes in the last 27 years [53]. In addition, NoroSurv, which is a global paediatric norovirus strain surveillance network for children <5 years of age hospitalised due to acute gastroenteritis reported GI.3 as the most prevalent GI genotype [61]. The dominance of clinically relevant, putative, and confirmed novel recombinant GI.3 viruses in our wastewater systems can be a mirror of viruses circulating in SA communities or it can be an early indicator of possible outbreaks.

The predominance of non-GII.4 noroviruses in SA wastewater was previously described. In that study, GII.2 viruses were predominant in wastewater from the Free State and Gauteng provinces between 2015 and 2016 [19]. Between 2016 and 2017, their dominance was seen in wastewater and a sewage-polluted river in Pretoria and Johannesburg, respectively [62]. In this study, GII.2 was the most frequently detected GII virus found in more than 50% of the amplicon pools; however, this trend is not unique to SA as data from GenBank also shows GII.2 strains to be among the top three norovirus genotypes reported globally between 1995 and 2019 [53]. An observation of concern made is that these viruses were found harbouring five different P-types, which include GII.P31 and co-dominant GII.P7 and GII.P16. The remaining two P-types, GII.P17, and GII.PNA together with GII.P7 were identified as putative novel GII.2 recombinants. Acquisition of these novel P-types by GII.2 is likely to escalate their capsid evolution, which might significantly increase their infectivity and prevalence potential [63]. Without clinical surveillance data, no conclusion can be drawn on the medical importance of these viruses. Norovirus GII.17’s prevalence in wastewater and sewage-polluted surface waters increased with the global emergence of the novel Kawasaki_2014 sub-variants 308 and 323. These viruses replaced GII.4 Sydney_2012 as the dominant strain causing norovirus-associated gastroenteritis in East Asian regions [64,65,66]. Other countries such as the USA [67], Italy [68], and Romania [69] only experienced the increase in GII.17-associated outbreaks and sporadic cases but not replacement of GII.4 Sydney_2012 as a dominant strain. Interestingly, GII.4 (median age = 1 year) and GII.17 (median age = 49 years) viruses have different age distributions and therefore, the age group targeted by clinical surveillance impacts the estimated genotype prevalence in a population. In SA, the shift in dominance was only observed in the wastewaters [19] and this is likely due to exclusion of diarrhoeal cases from individuals > 5 years by the majority of clinical surveillance studies. In addition, SA does not have a norovirus outbreak reporting system and therefore, GII.17 infections are likely missed and therefore underreported. This study revealed the circulation of the novel Kawasaki and original GII.17 variants in Pretoria. The latter formed a distinct cluster of strains that share a common ancestor with strain KC495674 that was previously identified in SA, and they are not closely related to any strain reported from other regions of the world (Figure 8). Norovirus GII.17 is not the only genotype with strains unique to SA; GII.4 Sydney_2012 strains clustered separately from four strains that could not be assigned to an existing and named variant. Analysis of complete VP1 nucleotide sequences of norovirus strains, especially those that are unique to specific parts of the world would assist in assigning them to a variant. The emerging GII.12[P16] recombinant was detected once in a raw sewage sample from January 2019 and during the same period, it was linked to norovirus-associated gastroenteritis outbreaks in Canada [70]. In October 2020, it caused a gastroenteritis outbreak in Brazil where the majority of cases were linked to the consumption of a local brand of ice pop [71]. Related strains were detected in the USA (MK754447), Spain (MT501819), and Japan (LC579431) between 2018 and 2020.

Clinical norovirus surveillance in SA is focused on children <5 years of age, hospitalised with gastroenteritis, which represents severe disease. During the study period the dominant genotype detected in this population was GII.4[P31] Sydney_2012 (unpublished data) which shared 98.8% nucleotide identity with the GII.4 strains detected in raw sewage. In a previous 5-year clinical norovirus surveillance study, GII genotype detection peaked from August to November every year [72,73]. In contrast, in this study norovirus GI and GII strains were detected in wastewater samples throughout the year with no distinct seasonality. The amplicon-based NGS approach was a limitation of the study, since this method can be biased to viruses that match the primers used; therefore, novel strains could be missed. With more resources, deeper sequencing using cDNA on more advanced Illumina platforms can provide more valuable data. Another limitation was disruption of sample collection by protests and COVID-19 pandemic lockdown restrictions.

## 5. Conclusions

Our analyses revealed a vast diversity of noroviruses circulating Pretoria, of which the majority were putative novel recombinants. Phylogenetic analyses showed the presence of uncommon strains that have not been detected worldwide and detected for the first time in SA. Furthermore, this study confirmed the emergence of four novel recombinants, including GII.12[P16]. Overall, this study contributed to the growing norovirus diversity data in SA and proved the usefulness of NGS in environmental surveillance. These data will assist the on-going vaccine and drug development studies, which rely on the continuous submission of norovirus sequences to public platforms such as GenBank.

## Figures and Tables

**Figure 1 viruses-14-02732-f001:**
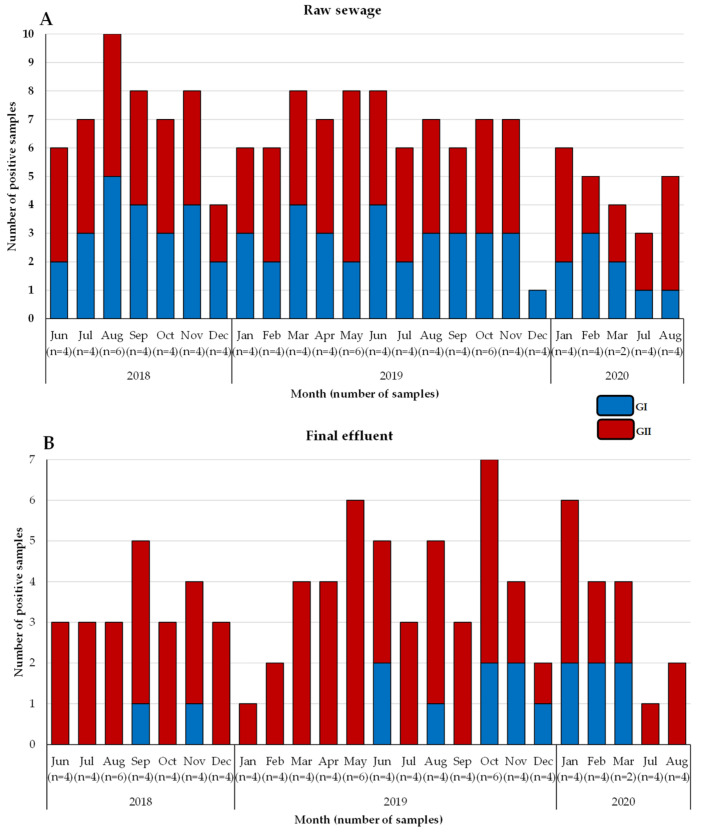
Distribution and prevalence of norovirus GI and GII in wastewater samples. Norovirus GI and GII detected in raw sewage (**A**) and final effluent (**B**) samples collected between June 2018 and August 2020. The numbers in parentheses indicate the number of samples collected in each month.

**Figure 2 viruses-14-02732-f002:**
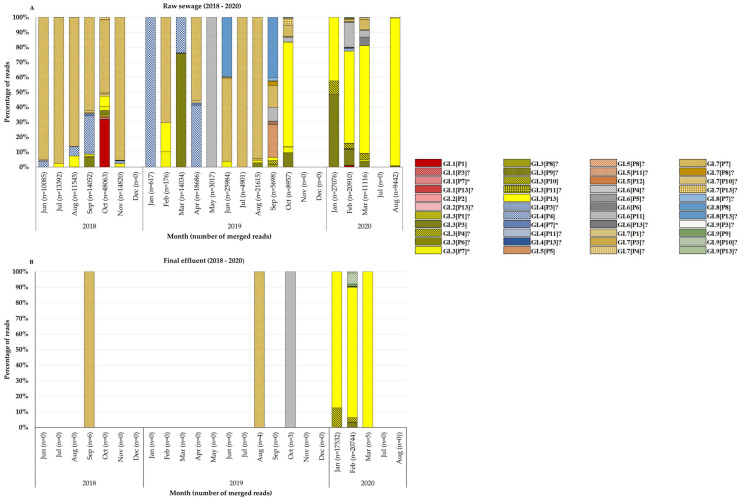
Norovirus GI molecular typing. Distribution and abundance of norovirus GI strains identified from (**A**) raw sewage and final effluent (**B**) samples between June 2018 and August 2020. The amplicons from different sampling sites were pooled equimolar based on wastewater sample type and month of collection. The numbers in parentheses indicate the number of merged reads for each month. The key shows known RdRp-capsid combinations, putative novel recombinants (?) and Sanger-sequencing confirmed novel recombinants (*).

**Figure 3 viruses-14-02732-f003:**
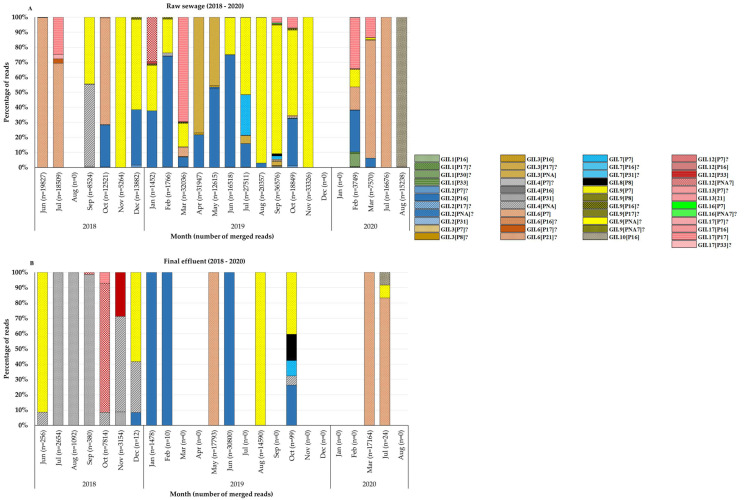
Norovirus GII molecular typing. Distribution and abundance of norovirus GII strains identified from (**A**) raw sewage and final effluent (**B**) samples between June 2018 and August 2020. The amplicons from different sampling sites were pooled equimolar based on wastewater sample type and month of collection. The numbers in parentheses indicate the number of merged reads for each month. The key shows known RdRp-capsid combinations and putative novel recombinants (?).

**Figure 4 viruses-14-02732-f004:**
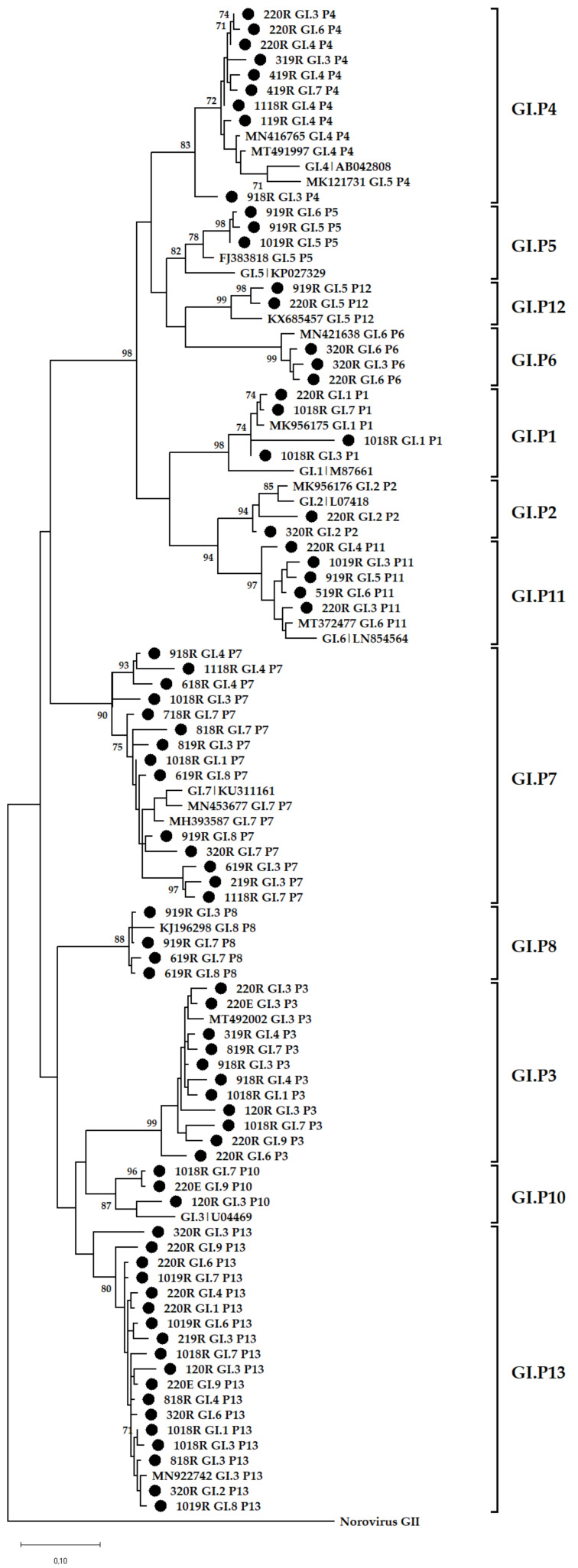
Phylogenetic analysis of the 265 bp at 3′-end of norovirus GI ORF1. Representative sequences identified in this study are indicated by black circles (●). All strain sequence names contain specific ID, genotype and P-type. Reference strains and the most closely related strains were downloaded from GenBank. The Maximum Likelihood phylogenetic trees were constructed using MEGA X software [46] and 1000 bootstrap replicates based on the Kimura 2-parameter model. The sequences were labelled according to month and year of detection, wastewater type, P-type and genotype, e.g., 618RGI.2[P2] = GI.2[P2] virus detected in raw sewage collected in June 2018.

**Figure 5 viruses-14-02732-f005:**
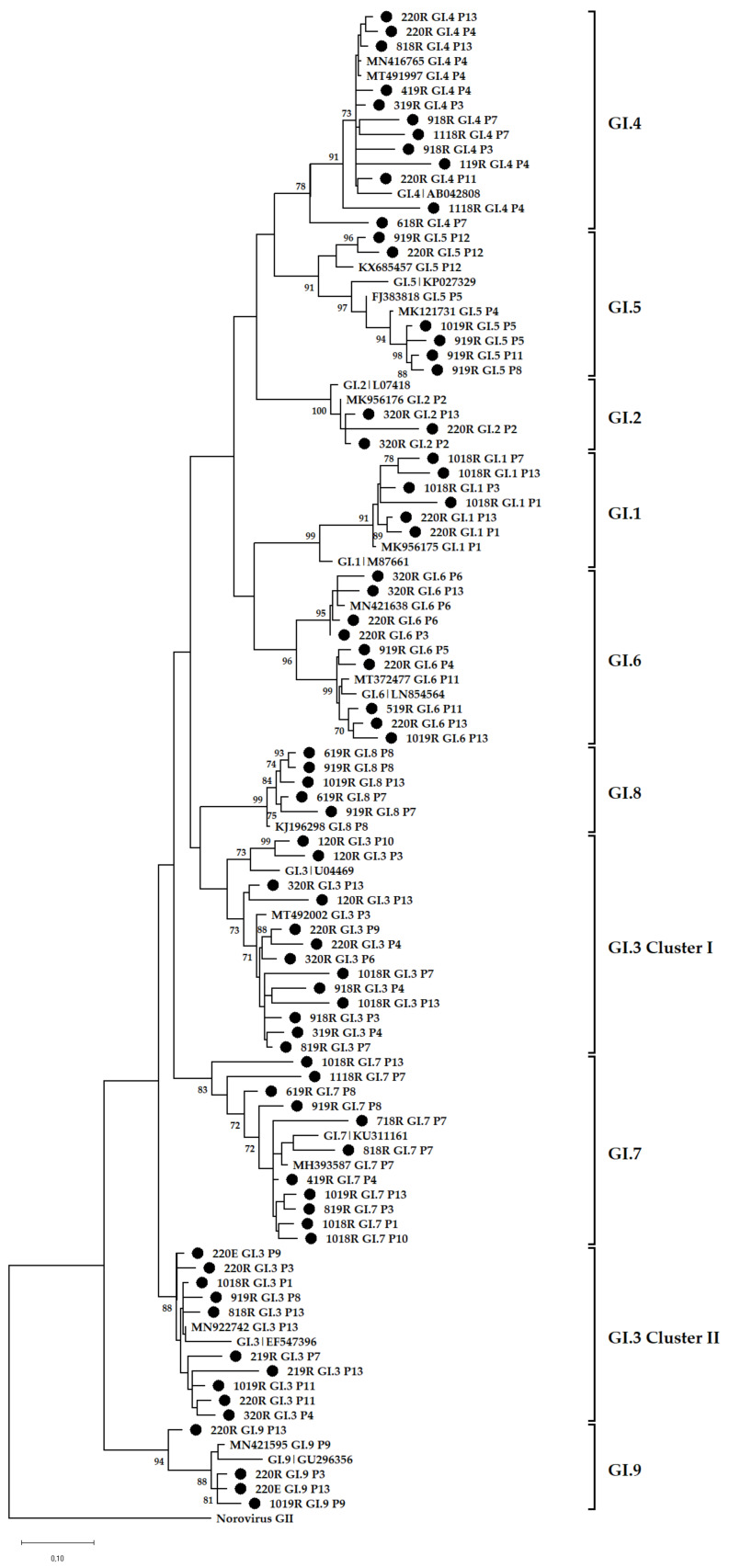
Phylogenetic analysis of the 314 bp at 5′-end of norovirus GI ORF2. Representative sequences identified in this study are indicated by black circles (●). All strain sequence names contain specific ID, genotype and P-type. Reference strains and the most closely related strains were downloaded from GenBank. The Maximum Likelihood phylogenetic trees were constructed using MEGA X software [46] and 1000 bootstrap replicates based on the Kimura 2-parameter model. The sequences were labelled according to month and year of detection, wastewater type, P-type and genotype, e.g., 618RGI.2[P2] = GI.2[P2] virus detected in raw sewage collected in June 2018.

**Figure 6 viruses-14-02732-f006:**
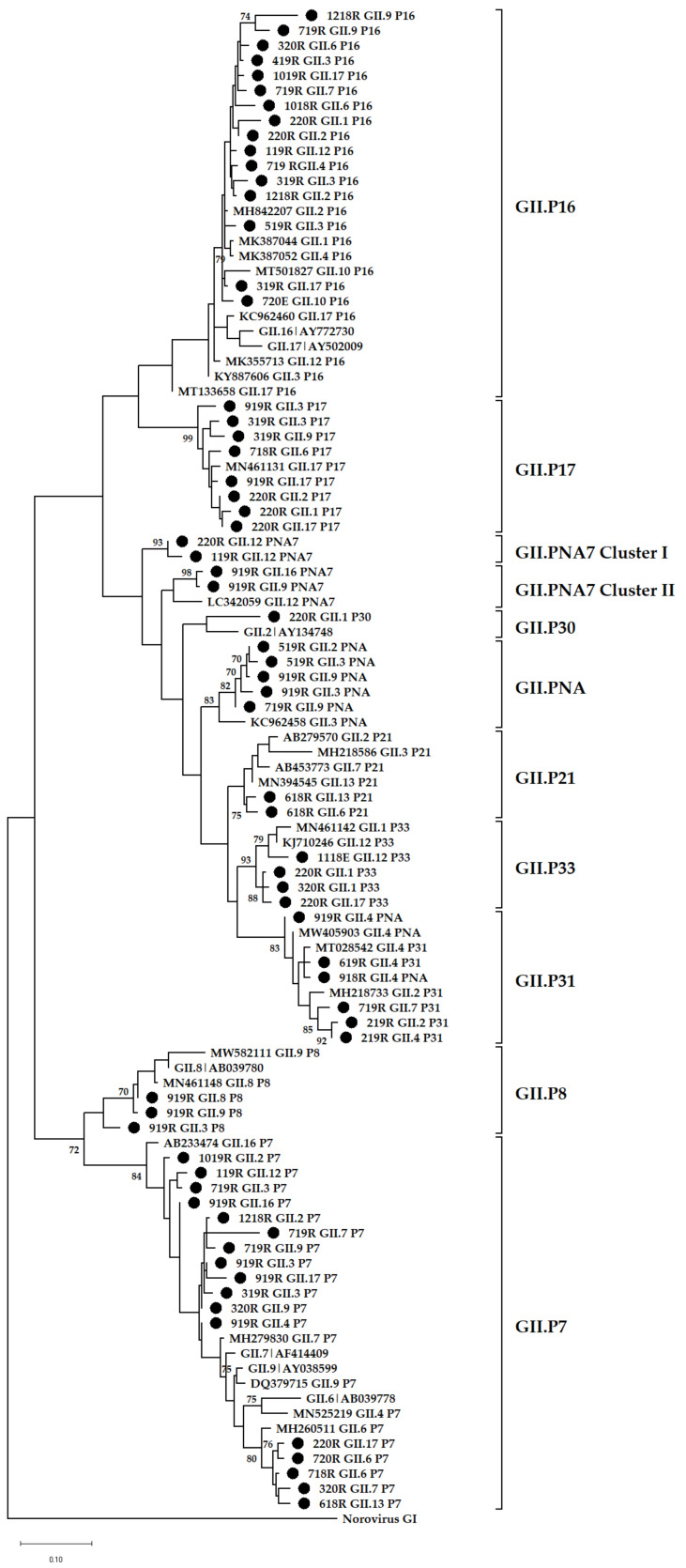
Phylogenetic analysis of the 265 bp at 3′-end of norovirus GII ORF1. Representative sequences identified in this study are indicated by black circles (●). All strain sequence names contain specific ID, genotype and P-type. Reference strains and the most closely related strains were downloaded from GenBank. The Maximum Likelihood phylogenetic trees were constructed using MEGA X software [46] and 1000 bootstrap replicates based on the Kimura 2-parameter model. The sequences were labelled according to month and year of detection, wastewater type, P-type and genotype, e.g., 618RGII.2[P2] = GII.2[P2] virus detected in raw sewage collected in June 2018.

**Figure 7 viruses-14-02732-f007:**
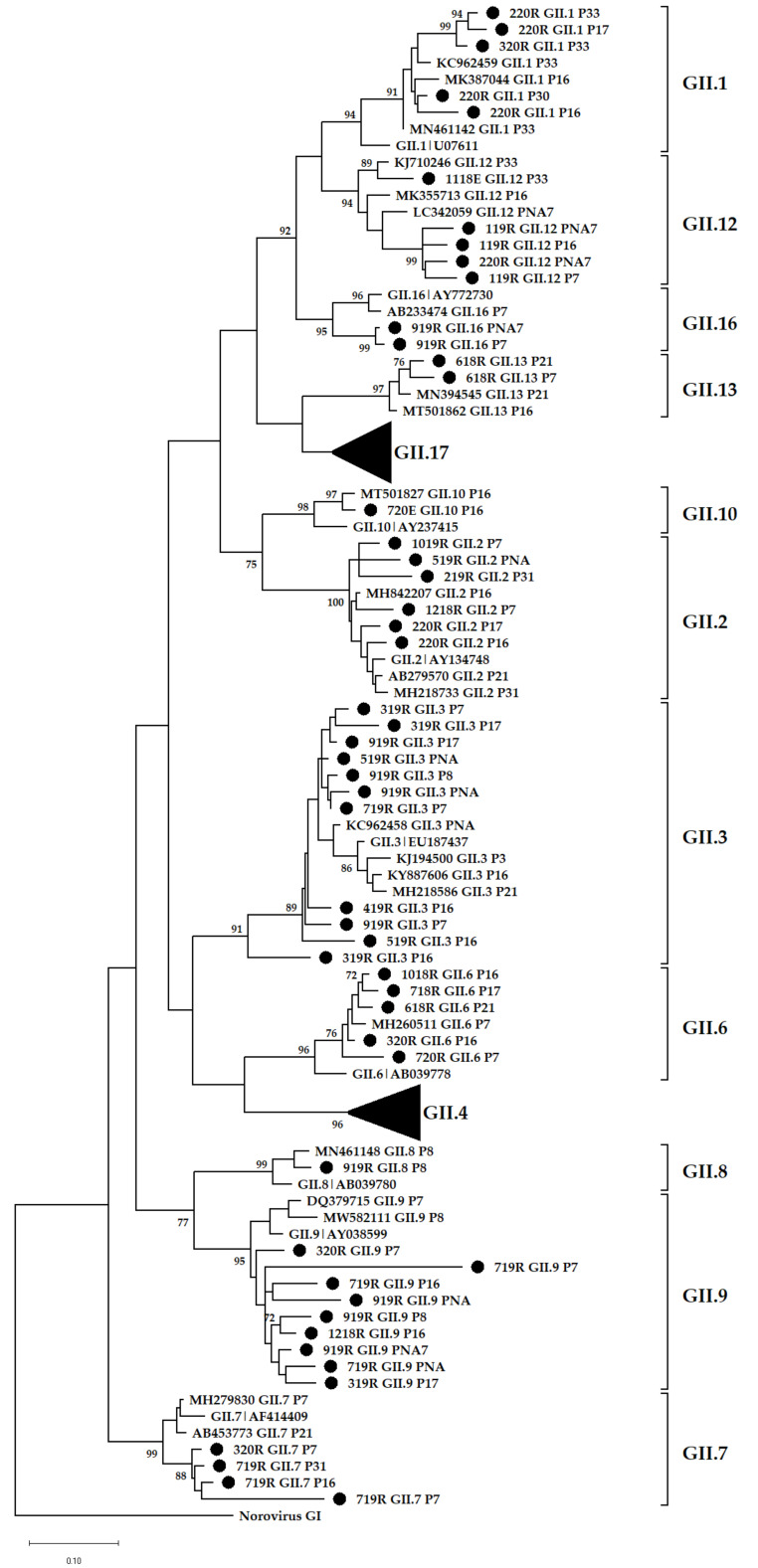
Phylogenetic analysis of the 305 bp at 5′-end of norovirus GII ORF2. Representative sequences identified in this study are indicated by black circles (●). All strain sequence names contain specific ID, genotype and P-type. Reference strains and the most closely related strains were downloaded from GenBank. The Maximum Likelihood phylogenetic trees were constructed using MEGA X software [46] and 1000 bootstrap replicates based on the Kimura 2-parameter model. The sequences were labelled according to month and year of detection, wastewater type, P-type and genotype, e.g., 618RGII.2[P2] = GII.2[P2] virus detected in raw sewage collected in June 2018.

**Figure 8 viruses-14-02732-f008:**
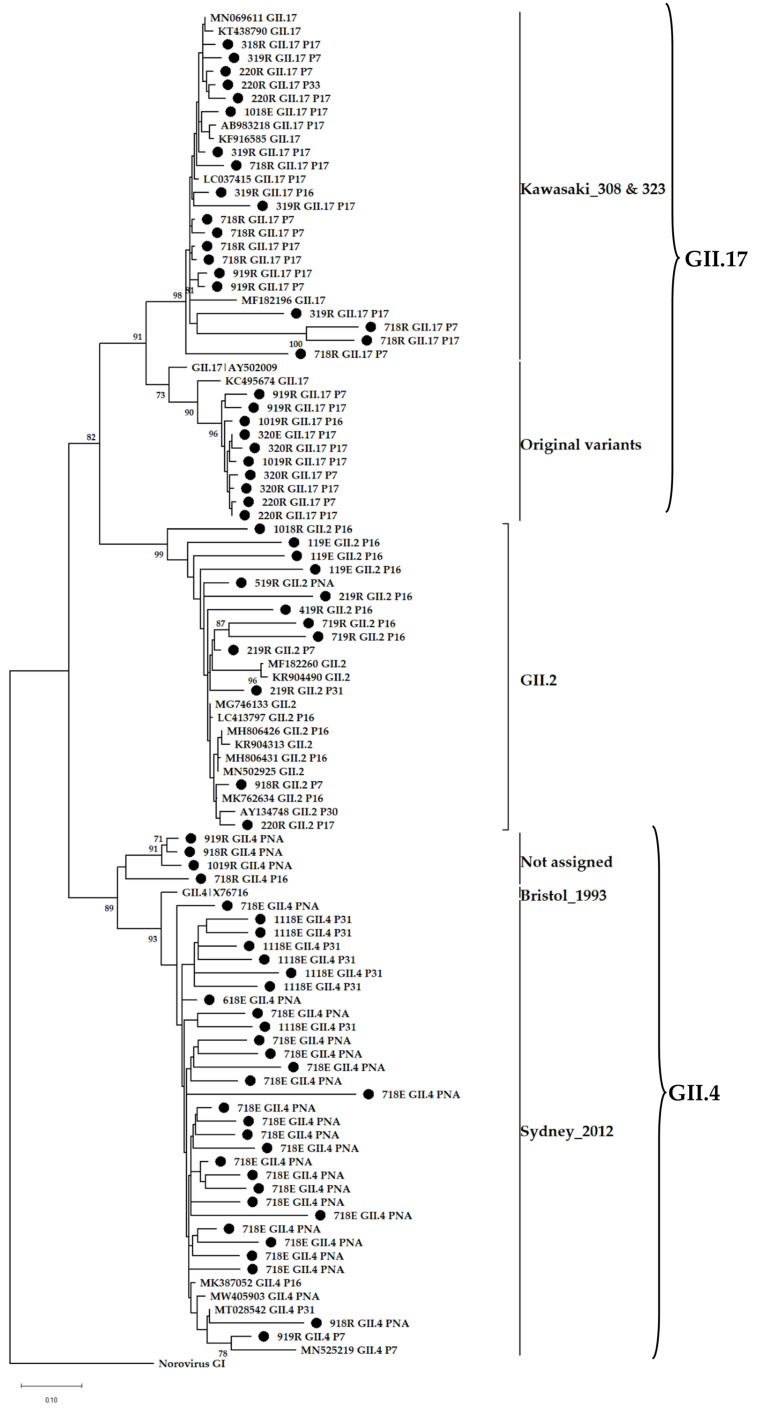
Phylogenetic analysis of the 302 bp at 5′-end of norovirus GII.2, GII.4 and GII.17 ORF2. Representative sequences identified in this study are indicated by black circles (●). All strain sequence names contain specific ID, genotype and P-type. Reference strains and the most closely related strains were downloaded from GenBank. The Maximum Likelihood phylogenetic tree was constructed using MEGA X software [46] and 1000 bootstrap replicates based on the Kimura 2-parameter model. The sequences were labelled according to month and year of detection, wastewater type, P-type and genotype, e.g., 1018RGII.2[P2] = GII.2[P2] virus detected in raw sewage collected in October 2018.

## Data Availability

The data presented in this study are available in the article and Appendix A.

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
