# Peer review of "Multiple Novel Human Norovirus Recombinants Identified in Wastewater in Pretoria, South Africa by Next-Generation Sequencing"

_viruses, 2022, doi:10.3390/v14122732_

Round 1
Reviewer 1 Report
The authors present a very good study on the genetic diversity of noroviruses in wastewater samples from 2 sewage treatment plants in Pretoria, South Africa. The results are clearly described and also of great interest for the readership of Viruses.
Main Point:
The authors describe the diversity of circulating noroviruses in wastewater samples. Here, the proportion of GI viruses detected in raw sewage samples is relatively high at 20%, compared to clinical samples in which a much lower proportion of GI viruses are often detected. The study would benefit greatly if comparative data from a clinical surveillance of circulating noroviruses in South Africa could be included in the study.
Minor Points:
- Line 5: Please check the enumeration of authors, has an author been forgotten as the "and" is the last word?
- Line 214: there is mention of a significant difference here, but a statistic is not described in the methods section. If significance was not tested, then delete accordingly, or describe significance test.
- Line 250: 12 GII.PNA sequences were detected. However, in the phylogenetic tree (Fig 6) only 4 GII.PNA7 and 5 GII.PNA (related to KC962458) sequences are shown. Which 3 other sequences were detected?
- Figure 2 and 3: The labels of the two axes are too small now not readable
Reviewer 2 Report
The manuscript entitled “Multiple novel human norovirus recombinants identified in wastewater in Pretoria, South Africa by next-generation sequencing” described the genotyping of noroviruses detected in raw sewage and final effluent wastewater during 2018-2020 in Pretoria, South Africa. The results are interesting and highly valuable, since the Authors detected a novel norovirus recombinants and applied well established methodology. The subject is important and interesting for a wide audience, leading to better understand norovirus epidemiology, possibly leading to improvement of preventive methods. The conclusions are fully supported by the obtained data. Language is very well, and everything is clear and easy to understand, except few sentences, which I asked to be rephrased. The study may be accepted after improvement or explanation of the following minor issues:
Line 104-106: Except this mengovirus extraction control, any other internal control of sample pre-processing have been used? How about the widely used for sewage surveillance PMMoV internal control, was it applicable in Your study? Why or why not? Would it be worth to spike the raw sewage sample with the known concentration of any virus to monitor the recovery of the virus during flocculation and other processing steps? Please, elaborate.
Line 115: why the value of 1% of extraction efficiency threshold has been applied?
Line 208: what was the frequency of sample collection? Were they collected in a given time intervals, for example every two weeks, every month, etc?? Or randomly collected during the whole study timeframe?
Lines 211-212: “Overall, noroviruses were detected in 81% (162/200) [89% (89/100) raw sewage and 73% (73/100) final effluent] of wastewater samples screened.” This sentence is quite unclear, please rephrase it.
Figure 1: Does the results stand in line with epidemiological data about gastroenteritis in Pretoria? Is there any correlation with climate conditions in Pretoria?
Figure 2&3: please improve the resolution of the figures, since it is hard to read the number of reads.
Lines 385-387: I agree with this statement, however please remember that for this Illumina MiSeq sequencing, you have used amplicons and this approach has some limitations, for example it can lead to the missing of some variants with mutations within primer binding sites, leading to lack of amplification. Please, write few words abouts the limitations of applied approach, and how they could be overcame. How about the flocculation method? Does it have any influence on the obtained results? How about the environmental RNA from destroyed Noroviruses, would it be possible to collect it through the used method?
Line 388: How this numbers relate to other similar studies, in other regions? Is 81% a high value?
Line 400-402: Please verify, whether you really mean the stability of genetic material or infectious virions, since it makes a big difference.
Lines 403-405: Was this difference in virus stability statistically significant?
Line 406: what is “uncompromised capsid”? I would rather use the “undamaged capsids” wording, or just “infective virions”, “complete virions”, “undamaged virions” etc.
Lines 406-409: the sentence is too long, please rephrase it.
Line 419-422: is there any alternative approach to obtain more valuable data regarding these recombinants? Would it be possible to perform deeper sequencing with more advanced Illumina platforms, without previous amplification step?
Round 2
Reviewer 1 Report
The authors did a good job of incorporating the criticisms into the present manuscript, which improved it. I have no other points that need editing. For this reason, I would recommend that the manuscript be accepted for publication in Viruses.